# A homogeneous bioluminescent immunoassay to probe cellular signaling pathway regulation

Byounghoon (Brian) Hwang[1]*, Laurie Engel[1], Said A. Goueli [1,2] & Hicham Zegzouti [1]*

Monitoring cellular signaling events can help better understand cell behavior in health and disease. Traditional immunoassays to study proteins involved in signaling can be tedious, require multiple steps, and are not easily adaptable to high throughput screening (HTS). Here, we describe a new immunoassay approach based on bioluminescent enzyme complementation. This immunoassay takes less than two hours to complete in a homogeneous "Add and Read" format and was successfully used to monitor multiple signaling pathways' activation through specific nodes of phosphorylation (e.g pIκBα, pAKT, and pSTAT3). We also tested deactivation of these pathways with small and large molecule inhibitors and obtained the expected pharmacology. This approach does not require cell engineering. Therefore, the phosphorylation of an endogenous substrate is detected in any cell type. Our results demonstrate that this technology can be broadly adapted to streamline the analysis of signaling pathways of interest or the identification of pathway specific inhibitors.

[1] R&D Department, Promega Corporation, 2800 Woods Hollow Road, Madison, WI 53711, USA. [2] Department of Pathology and Laboratory Medicine, University of Wisconsin School of Medicine and Public Health, Madison, WI, USA. *email: brian.hwang@promega.com; hicham.zegzouti@promega.com

Post-translational modification (PTM) is a key process to regulate protein function by modulating their conformation, stability, their affinity for interacting proteins, and subcellular localization. The functional regulation of proteins by PTM is critical for cellular signaling, which provides a rapid response to dynamic changes in intracellular or extracellular conditions. Dysfunction of this process often leads to pathological conditions, such as cancer, or metabolic and inflammatory diseases. Post-translational modifications include phosphorylation, ubiquitination, sumoylation, acetylation, glycosylation, methylation, and hydroxylation. Phosphorylation is one of the most well-studied PTMs due to its critical role in cellular signal pathway regulation. In these pathways, phosphorylation of specific proteins by specific kinases constitute important nodes by which the signal is transduced from upstream activation events to downstream cellular responses.

Cellular protein phosphorylation has been routinely monitored by various methods such as western blot analysis, ELISA, and mass spectrophotometry. In these methods, a sufficiently high number of cells is stimulated so that the native phosphoprotein level can be readily detected[1]. In some cases, lengthy sample preparation steps can compromise accuracy, especially in a rapid response pathway where dephosphorylation may occur during sample preparation (e.g. NF-κB activation by TNFα[2]). In other cases, detection of a low abundance phosphoprotein is facilitated by enrichment through immunoprecipitation. This approach may include overexpression of the target with a tag (e.g. myc) that enables immunoprecipitation from a relatively small cell number. However, overexpressed proteins may function differently from the native protein possibly due to improper subcellular localization or unbalanced stoichiometry with binding partners[3]. Recently, CRISPR/Cas9 techniques have been developed to express endogenous level of proteins by editing the cell genome and introducing small tags in frame with the target protein[4–6]. Although some of these methods can be sensitive to detect native PTM levels, they can be tedious, require multiple washing steps (non-homogeneous), or the necessary cell engineering is a requisite upfront hurdle for assay design. Some homogeneous cell-based PTM assays were also developed using fluorescence or energy transfer (e.g. FRET)[7,8]. However, they are vulnerable to false hits due to fluorescence interference, signal quenching, light sensitivity, and require special instruments for detection. Therefore, there is a need for a simpler approach with the necessary sensitivity to detect PTM of native proteins.

In the present study, we describe simple homogeneous cell-based assays to detect both the phosphorylation and total level of native target proteins in unmodified cells using immunodetection combined with protein-subunit complementation based on NanoLuc luminescence. NanoLuc is an engineered 19 kDa luciferase enzyme derived from the deep-sea luminous shrimp *Oplophorus gracilirostris*, that is relatively stable and produces very bright and sustained bioluminescence[9]. Recently, NanoLuc Binary Technology (NanoBiT) has been developed as an adaptation of Nanoluc luciferase[10]. NanoBiT is a structural complementation reporter ideal for protein:protein interaction (PPI) studies[11–13]. The NanoBiT system is composed of two subunits, Large BiT (LgBiT; 18 kDa) and Small BiT (SmBiT; 11 amino acid peptide), that can be expressed as fusions or chemically conjugated to target proteins of interest. The LgBiT and SmBiT subunits are stable and have minimal self-association because the SmBiT peptide associates with the LgBiT protein with relatively low affinity (KD = 190 μM)[10]. Therefore, only when two target proteins tagged with these subunits interact, the subunits come together to form an active enzyme and generate a bright luminescent signal in the presence of its substrate furimazine. In this new immunoassay, the NanoBiT subunits are conjugated to an anti-mouse and an anti-rabbit secondary antibody (NanoBiT detecting antibodies), which recognize two primary antibodies to a single protein, one to a phosphorylated and the other to a non-phosphorylated epitope, or to two non-phosphorylated epitopes. In this way the system detects the phosphorylated or total levels of specific protein targets in a cell lysate.

The bioluminescent cell-based immunoassay presented here has advantages over current technologies, including the fact that it doesn't require cell engineering, which enables target detection in any cell type including primary cells. The assay only requires a simple luminometer, it is homogeneous (does not require cell collecting or washing steps), and its simple "add and read" format enables detection of quick response signaling events with a relatively small number of cells. Also, the system is modular requiring only a pair of primary antibodies against a target to pair with the NanoBiT detecting antibodies. The utility of the approach for measuring signaling pathway regulation was demonstrated by targeting specific phosphoproteins as nodes of pathway activation. Examples include IκBα or p65, AKT, and STAT3 as respective phosphoprotein nodes of the NF-κB, AKT, and JAK/STAT pathways. In each case the expected biological response was observed as a change in the abundance or phosphorylation level of the target protein. We used the assays to determine IC$_{50}$ values for small or large molecule pathway inhibitors in a sensitive and selective way. Our results demonstrate that this bioluminescent technology can be broadly adapted for signaling pathway analysis. These applications will help scientists identify and validate novel pathway specific drug therapies for the treatment of various diseases.

## Results

**NanoBiT cell-based immunoassay principle and formats**. In multi-well plates, stimulated cells containing the target protein are lysed and antibody solution is added to the cell lysate before the mixture is incubated at room temperature (Fig. 1). This antibody solution contains the NanoBiT detecting antibodies and two primary antibodies (one raised in rabbit and one in mouse species). The primary antibodies recognize separate epitopes on a single target protein. For measuring target protein phosphorylation, one of the primary antibodies is selective for a phosphorylated epitope. Binding of the primary/detecting antibody complexes to their corresponding epitopes brings NanoBiT subunits into proximity to form an active NanoLuc luciferase that makes light in proportion to the amount of the target protein. When the primary antibody pair includes a phospho-specific antibody, the luminescence reflects the level of the target protein phosphorylation (Fig. 1a). The assays were typically completed in <2 h (Fig. 1b). This two-epitope/single target approach greatly enhances target selectivity for both phospho- and non-phosphoproteins because non-specific primary antibody binding has a very low probability of mediating productive NanoBiT complementation via proximity binding.

It was important to demonstrate that the NanoBiT-labeled secondary antibodies have high species selectivity for their respective primary antibody targets. To do that, mouse or rabbit IgG antibodies were used as analytes and detected with either two NanoBiT-labeled anti-mouse or anti-rabbit secondary antibodies where one is labeled with SmBiT and the other labeled with the LgBiT fragments. The NanoBiT-labeled secondary detecting antibodies show excellent selectivity for their respective targets (Supplementary Fig. 1).

Non-covalent protein:protein interaction is the basis for NanoBiT complementation and the complex dissociation rates are accelerated with increasing concentrations of the detergents typically used to create cell lysates. To achieve

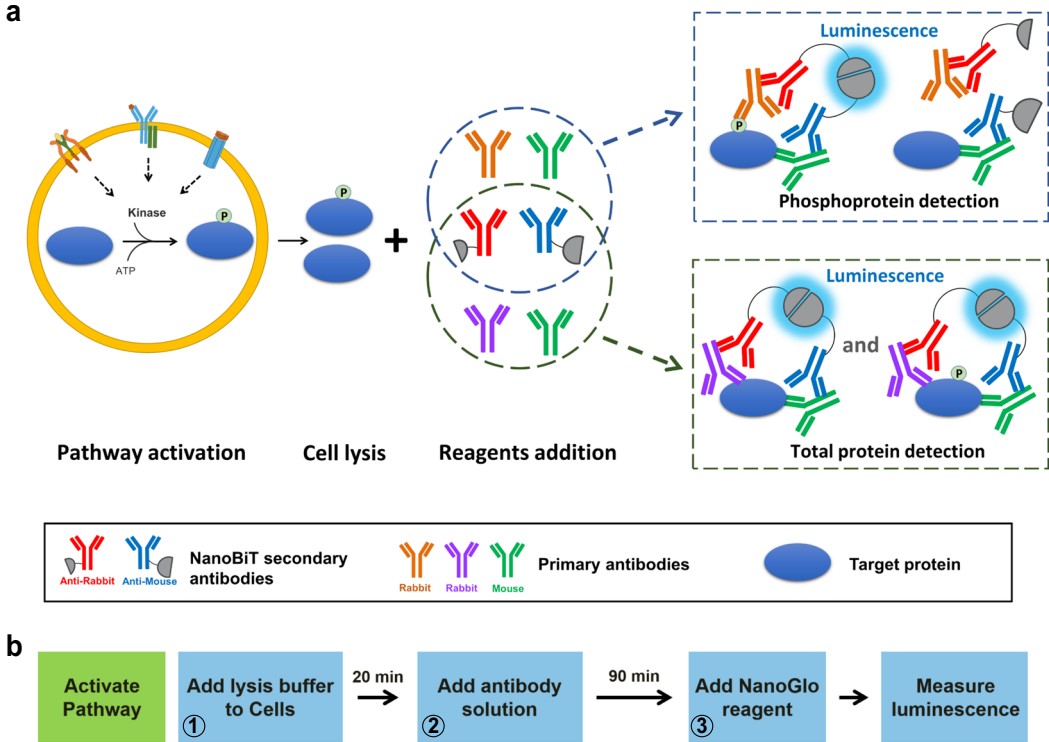

**Fig. 1 NanoBiT cell-based immunoassay. a** Principle of NanoBiT cell-based immunoassay. Phosphorylated or total target proteins in lysed cells after stimulation are recognized by each primary antibody pair. The NanoBiT conjugated secondary antibodies then recognize their cognate primary antibodies, resulting in close proximity of the NanoBiT subunits to form a functional enzyme that generates bright luminescence. **b** Workflow of NanoBiT cell-based immunoassay. A homogeneous and simple "Add and Read" cell-based assay format.

complete cell lysis while preserving light output (Fig. 1b, step 1), the lysis buffer is diluted when the antibody solution is added (Fig. 1b, step 2). We screened multiple detergents at different concentrations for their compatibility with the NanoBiT complementation and found that lysis in the presence of 0.02% digitonin is the most compatible as it adequately breaks the cells and cellular compartments (verified under microscope during optimization of digitonin concentration and the incubation time) to allow availability of all the cellular proteins to antibodies added in the second step of the assay. The addition pattern for optimal performance follows the 2:2:1 ratio of the cell lysate:antibody solution:Nano-Glo reagent, with volume ratios of 50:50:25 µl used for 96-well plates and volume ratios of 10:10:5 µl used for 384-well plates. Following this pattern, the final digitonin concentration is 0.008% resulting in no interference with complementation and NanoBiT light output.

The immunoassay described here is an end point homogeneous assay which means the lysis, the antibody binding to the target and the luminescence generation steps all occur in solution without liquid transfer or washing steps. At the end of the 20 min digitonin lysis step, the cells in the well are lysed completely to create a homogeneous lysate. At this step, cellular activities such as signal transduction and protein expression are halted. When the antibody mix and the luciferase detection reagent are added to the lysate, the luminescent signal detected represents an accurate measure of the state of the phosphorylation of the cells before lysis.

**Optimizing the immunoassay for total and phospho-proteins.** To develop an immunoassay to detect any target protein, the quality and availability of primary antibodies are of upmost importance. In order to demonstrate the feasibility of the

NanoBiT cell-based immunoassay for detecting phosphorylation of a target protein, we chose the NF-κB signaling pathway node IκBα as a target protein model because it is an important signaling node that goes through multiple processes upon pathway activation. The assay was used to detect IκBα phosphorylation by the IKK complex and to follow its proteasomal degradation after pathway activation[14–16]. Moreover, conventional methods to detect IκBα phosphorylation such as western blot analysis are challenging due to its immediate degradation after phosphorylation and ubiquitination[2]. The first step to optimize the assay is to select an optimal primary antibody pair and determine the optimal antibody concentration. We screened mouse/rabbit pairs of commercially available antibodies comprised of an anti-phospho-IκBα (Ser32) antibody and an antibody, which recognizes a different epitope on the protein. Three pairs made of two anti-p-IκBα and three anti-IκBα antibody combinations were titrated in a concentration matrix and used to detect phospho-IκBα in MCF-7 cells treated with TNFα. The same antibody combinations were also used to detect the basal level of p-IκBα in untreated cells and determine the fold increase in phosphorylation detection upon the pathway activation. All antibody pairs generated light above background (samples in which only NanoBiT detecting antibodies were used without the primary antibodies) at a certain concentration, but one pair (rabbit antibody #1/mouse antibody #1) generated the highest luminescent signals and signal fold change between treated and untreated cells at several primary antibody concentrations (Fig. 2a). The primary antibody pair #1 at 150 ng per ml (1 nM) was selected for further study because an optimal fold change of IκBα phosphorylation in response to TNFα treatment and a reasonable light output were obtained. We then optimized the incubation time of the primary and NanoBiT detecting antibody mixture with the cell lysate. The mix was incubated for different times before the addition of the

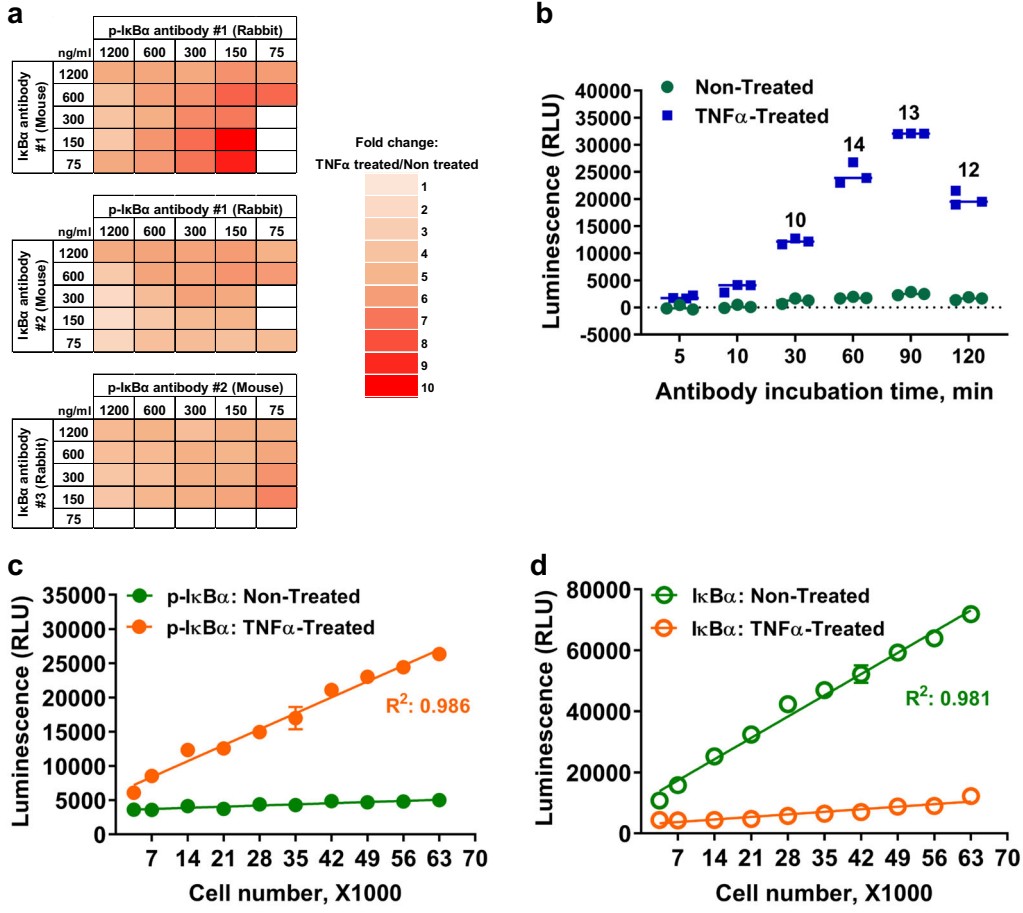

**Fig. 2 Optimization of the NanoBiT cell-based immunoassay with detection of IκBα phosphorylation or degradation. a** In all, 50,000 seeded MCF-7 cells were untreated or treated with TNFα (50 ng per ml, 10 min). IκBα phosphorylation was measured by NanoBiT cell-based immunoassay using the indicated antibody pairs at the indicated concentrations in the reaction after addition of antibodies. Fold changes (TNFα-Treated/Non-Treated) were calculated and represented in a heat map style. **b** In all, 50,000 seeded MCF-7 cells were pretreated with MG132 (20 μM, 30 min) and then untreated or treated with TNFα (50 ng per ml, 30 min). IκBα phosphorylation was measured with various antibody incubation times (5, 10, 30, 60, 90, and 120 min) before the Nano-Glo reagent was added. Fold changes (TNFα-Treated/Non-Treated) were calculated for each time point and are shown above each pair of bars. **c** Various number of MCF-7 cells (from 3500 to 63,000) were seeded and incubated at 37 °C in a 5% $CO_2$ overnight. The cells were pretreated with MG132 (20 μM, 1 h) and were untreated or treated with TNFα (10 ng per ml, 30 min) before IκBα phosphorylation was measured by NanoBiT cell-based immunoassay as described in Material and Methods section. **d** Various numbers of MCF-7 cells (from 3500 to 63,000) were seeded and incubated at 37 °C in a 5% $CO_2$ overnight. The cells were untreated or treated with TNFα (10 ng per ml, 30 min). Total IκBα level was measured by NanoBiT cell-based immunoassay as described in Material and Methods section. For **b–d** results are presented as means ± S.E.M ($n = 3$ technical replicates, the data are representative of two or more experiments).

Nano-Glo detection reagent and luminescence measurement. The increase of IκBα phosphorylation in TNFα-treated cells compared to non-treated cells was detected with all incubation times suggesting that at least a proportion of the antibodies bind fast to the target (Fig. 2b). However, maximal luminescence likely reflecting equilibrium binding was not reached until about 90 min. Nevertheless, the detected phosphorylation fold change (TNFα treated over non-treated) at different time points were similar as long as the antibodies were incubated with cell lysates longer than 30 min at room temperature (Fig. 2b). It should be noted based on the detection of other targets we studied that optimal antibody incubation time may depend on the affinity of the primary antibodies. However, incubation time for the antibody mix of 60 to 90 min was optimal for all targets that we have tested.

To determine an optimal cell number for the assay, phosphorylated IκBα was measured in various numbers of MCF-7 cells after TNFα treatment. In the absence of stimulation, the phospho-IκBα signal was at the no primary antibody background level and did not change as the cell number increased

suggesting that a basal level of IκBα phosphorylation was either absent or below the detection limit of the assay. In contrast, upon TNFα treatment, the phospho-IκBα signal increased with increasing cell number in a linear fashion (Fig. 2c). Similarly, by replacing the anti-phospho IκBα antibody with an anti-IκBα antibody that recognizes a different epitope of the protein, we were able to detect total IκBα protein in various numbers of MCF-7 cells. The assay showed that the level of total IκBα increased with increasing cell number in a linear fashion in the absence of TNFα treatment while the protein amounts were reduced with TNFα treatment due to its degradation (Fig. 2d). Changes in phosphorylation or degradation of IκBα in response to TNFα treatment could be observed with this assay using as low as 3500 cells with a linear range of detection between 3500 and 60,000 cells per well (Fig. 2c, d).

To assess the specificity of IκBα detection using the NanoBiT immunoassay, 50,000 cells per well, as an optimal condition, was used to measure phosphorylation and degradation of IκBα in response to specific known NF-κB pathway modulators. As

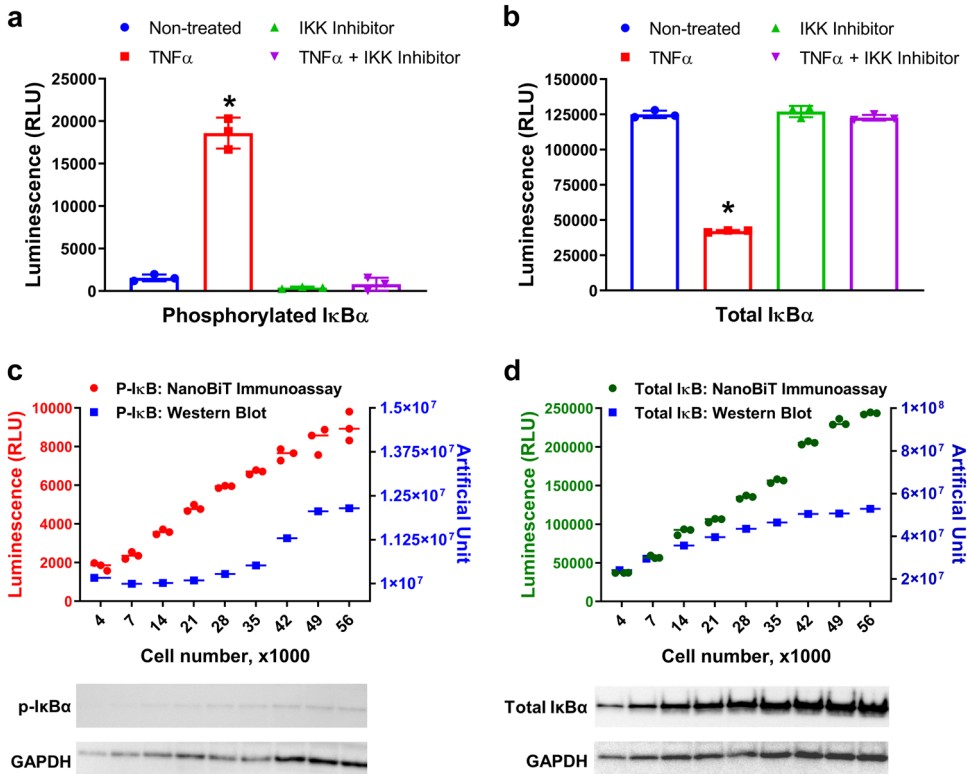

**Fig. 3 Validation of NanoBiT cell-based immunoassay detection of phosphorylated and total IκBα. a, b** In all, 50,000 seeded MCF-7 cells were untreated or pretreated with IKK16 (10 μM, 1 h) and then untreated or treated with TNFα (50 ng per ml, 10 min). Phosphorylated (**a**) or total IκBα (**b**) levels were measured by NanoBiT cell-based immunoassay using antibody pairs as described in Material and Methods. Results are presented as means ± S.E.M. ($n = 3$ technical replicates, the data are representative of two or more experiments). *$P < 0.001$ relative to Non-treated. **c** Various numbers of MCF-7 cells (from 3500 to 56,000) were seeded and incubated at 37 °C in a 5% $CO_2$ overnight. The cells were pretreated with MG132 (20 μM, 1 h) and then untreated or treated with TNFα (50 ng per ml, 30 min). IκBα phosphorylation was measured by NanoBiT cell-based immunoassay or Western blot analysis using the same primary antibodies. GAPDH blot served as a loading control. **d** Similar to **c** except cells were untreated or treated with only TNFα (50 ng per ml, 30 min). Total IκBα level was measured by NanoBiT cell-based immunoassay or western blot analysis using the same primary antibodies. GAPDH blot served as a loading control. Results for NanoBiT immunoassay in **c**, **d** are presented as means ± S.E.M. ($n = 3$ technical replicates, the data are representative of two or more experiments). Uncropped blots for **c**, **d** can be found in Supplementary Fig. 4.

expected, the phosphorylation of IκBα on serine 32 was increased after TNFα treatment and this phosphorylation was suppressed by IKK16, an IKK complex selective inhibitor (Fig. 3a). The amount of total IκBα protein also decreased due to its degradation after TNFα treatment and this reduction was abolished with IKK16 treatment (Fig. 3b). Taken together, these results show that NanoBiT immunoassay reagents combined with an IκBα primary antibody pair can, in a simple way reveal the predicted biology of NF-κB signaling pathway upon TNFα treatment.

To validate the data obtained using the bioluminescent assay in comparison to standard methods, we measured the level of phospho and total IκBα using the bioluminescent immunoassay compared to western blotting method. Different numbers of MCF-7 cells were seeded in 96-well plates in duplicate and treated or not with TNFα to detect phospho or total IκBα. One plate of each treatment was processed using the bioluminescent immunoassay and the other plates were subjected to a standard western blot protocol. Although both methods detected phospho and total IκBα in lysates from different cell numbers (Fig. 3c, d), the bioluminescent immunoassay was simpler and quicker, taking around 2 h from cell lysis to reading the luminescence. In contrast, western blot analysis took about 18 h because of the different incubations and washing steps (Supplementary Fig. 2). Another advantage is that the bioluminescent assay is more quantitative and has a broader linear response across all cell

numbers tested here. The western blot data was difficult to quantify and the detection of a low level of IκBα phosphorylation required longer western membrane exposure to only detect a faint band while the detection of total IκBα saturates at the higher cell numbers. These results show that phosphorylation or degradation of IκBα can be measured using the NanoBiT cell-based immunoassay in small number of cells and in a more quantitative way.

**Deciphering NF-κB pathway activation through IκBα node.** Some signaling pathways, such as the NF-κB pathway, are activated quickly in response to diverse stimuli such as TNFα or IL-1β. To study a rapid response pathway effectively, our simple homogeneous assay is ideal because it eliminates time consuming steps required in many other assay formats (Supplementary Fig. 2). These advantages allowed us to detect phosphorylation and degradation of IκBα in response to TNFα treatment at various and early time points. Change in IκBα phosphorylation was detected as early as 4 min and the phosphorylation signal started decreasing after 10 min of TNFα treatment (Fig. 4a). Since lysis buffer is directly added to break the stimulated cells without collecting them, the stimulation time in the NanoBiT cell-based immunoassay is closer to "real" stimulation time than other methods which require lengthy sample preparation that could extend dramatically the treatment time.

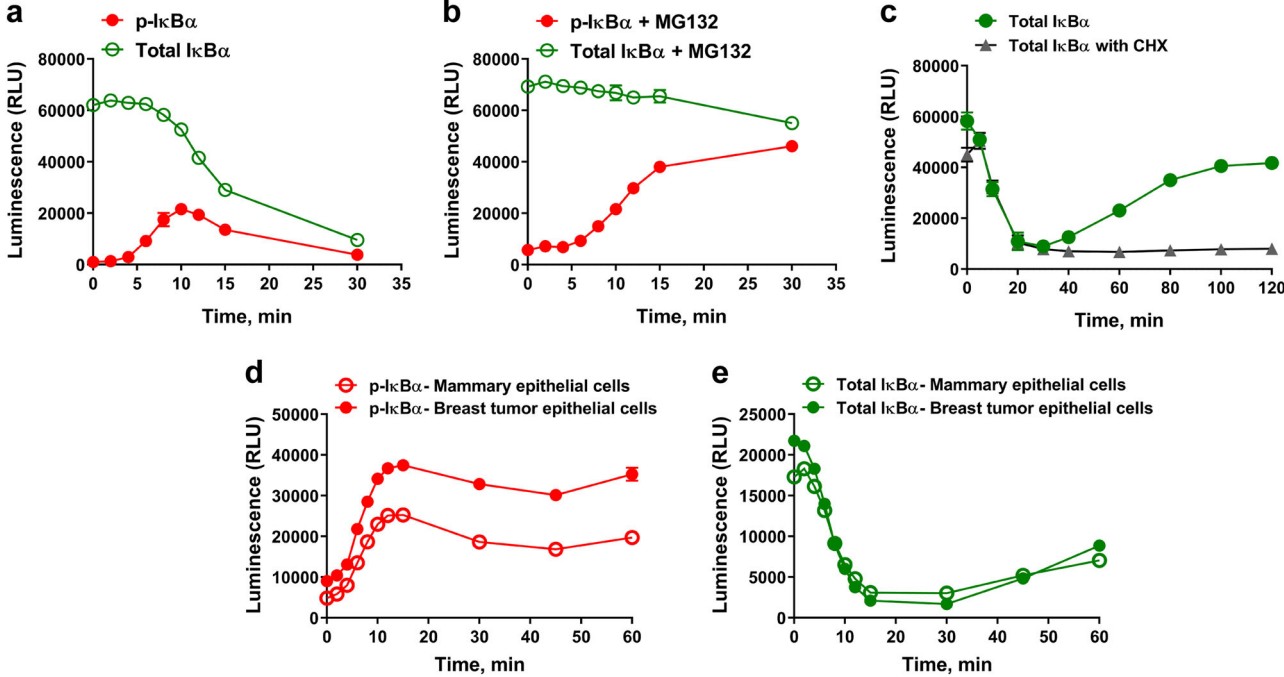

**Fig. 4 Deciphering NF-κB pathway activation by studying phosphorylation and degradation of IκBα.** In all, 50,000 seeded MCF-7 cells were untreated or treated with **a** TNFα (50 ng per ml) or **b** pretreated with MG132 (20 μM, 1 h) then TNFα (50 ng per ml) for various time points (2, 4, 6, 8, 10, 12, 15, and 30 min). Phosphorylated or total IκBα levels were measured by the corresponding NanoBiT cell-based immunoassay. **c** In all, 50,000 seeded MCF-7 cells were untreated or treated with TNFα (50 ng per ml) for a longer time course with Cycloheximide (20 μg per ml) added or not after the 30 min time point. Total IκBα level was measured by NanoBiT cell-based immunoassay. **d, e** Detection of phosphorylation and degradation of IκBα in human primary cells. **d** In all, 50,000 seeded human primary mammary epithelial cells or human breast tumor epithelial cells were pretreated with MG132 (20 μM, 1 h) before the cells were untreated or treated with TNFα (50 ng per ml) for various time points (2, 4, 6, 8, 10, 12, 15, 30, 45, and 60 min). Phosphorylation of IκBα was measured by the corresponding NanoBiT cell-based immunoassay. **e**, like **d**, except cells were treated or not with TNFα (50 ng per ml) in absence of MG132 for the various time points. Degradation of IκBα was measured by NanoBiT cell-based total IκBα immunoassay. Results are presented as means ± S. E.M. (n = 3 or four technical replicates, the data are representative of two or more experiments).

The decreased IκBα phosphorylation signal after 10 min of TNFα treatment may be due to immediate degradation of the phosphorylated IκBα. To test this, we also measured total IκBα level to assess its TNFα-mediated degradation. Interestingly, decreasing amounts of total IκBα protein were detected starting around 10 min after TNFα treatment confirming the decreased phosphorylation signal is due to degradation of total IκBα (Fig. 4a). Moreover, this suggests that there is an accumulation of the phospho form of IκBα for 10 min before the proteasome machinery starts degrading the protein and this takes another 20 min to complete, leaving only a residual level of IκBα. Since phospho-IκBα is the form targeted for degradation, and because the assay to detect total IκBα measures both phosphorylated and non-phosphorylated forms, the majority of IκBα is eliminated by 30 min, it suggests that most of the IκBα pool was phosphorylated during this time frame. In order to decipher the timing of phosphorylation and to confirm the IκBα degradation dependence on phosphorylation and their relative rates during the 30 min time course, MCF-7 cells were pretreated with the proteasome inhibitor MG132 prior to TNFα treatment. MG132 inhibited the degradation of IκBα while its phosphorylation signal kept accumulating for 15 min after TNFα treatment before stabilizing (Fig. 4b). This suggests that in the absence of MG132, IκBα phosphorylation is faster than its degradation up to 10 min but does not reach maximum phosphorylation before degradation dominates and depletes the entire pool. (Fig. 4a, b).

A regulatory mechanism of NF-κB signaling involves de novo IκBα expression[17]. IκBα is a target gene of the NF-κB signal transduction pathway[18,19], which turns off NF-κB activation by reforming the complex with NF-κB subunits that sequester them in the cytoplasm and terminate NF-κB mediated transcription[20]. We showed that TNFα treatment caused maximal IκBα degradation by 30 min and then extended the time course to 2 h of TNFα treatment to see if newly synthesized IκBα appears. In this case, after an initial 30-minute decline, IκBα protein started to accumulate again to almost starting levels (Fig. 4c). To confirm that this newly detected IκBα is a result of de novo transcription/translation of IκBα gene, the MCF-7 cells were treated with TNFα in the presence of cycloheximide for the same extended time to block protein synthesis and total IκBα was measured. In the presence of cycloheximide, IκBα was not detected after the degradation phase, suggesting the increase after 30 min without cycloheximide is newly synthesized IκBα protein (Fig. 4c) and that is consistent with the known NF-κB biology[2,17]. Furthermore, treating the cells with another cytokine, IL-1β which is also known to activate the NF-κB pathway showed similar results where IκBα is phosphorylated and degraded in a timely manner (Supplementary Fig. 3). However, while there was a similar pattern of IκBα degradation and phosphorylation in response to both cytokines, IL-1β stimulated a higher fold change in IκBα phosphorylation that reached a maximum slightly faster than TNFα in MCF-7 cells. This is consistent with the fact that these cytokines stimulate different pathways with the common downstream node, IκBα (Supplementary Fig. 3a).

In addition to cell lines, NanoBiT immunoassays were also applied to primary cells. We tested human breast tumor epithelial cells compared to normal mammary epithelial cells by measuring the level of total and phospho-IκBα plus and minus TNFα

**Table 1 List of protein targets whose phosphorylated and/or total level detected with NanoBiT immunoassay.**

| | Protein | Target | Cell lines | Stimulation | Inhibition |
|---|---|---|---|---|---|
| 1 | IκBα | Phospho-Ser32 | MCF-7, Hela, Ramos, human mammary | TNFα, IL-1β | IKK16 |
| | | Total[a] | primary, human breast cancer primary | TNFα, IL-1β | IKK16 |
| 2 | NF-κB p65 | Phospho-Ser536 | MCF-7 | TNFα, IL-1β | IKK16 |
| | | Total | MCF-7 | N/A | N/A |
| 3 | AKT | Phospho-Ser473 | MCF-7 | Insulin, IGF-1 | LY294002 |
| | | Total | MCF-7 | N/A | N/A |
| | | Phospho-T308 | MCF-7 | Insulin | LY294002 |
| 4 | STAT3 | Phospho-Tyr705 | A431 | IL-6 | Ruxolitinib, Tocilizumab, human anti IL-6 antibody |
| | | Total | A431 | N/A | N/A |
| 5 | BTK | Phospho-Tyr223 | Ramos-RA1 | Pervanadate | Ibrutinib |
| 6 | Rb protein | Phospho-Ser807/811 | MCF-7 | 17β-Estradiol, EGF | Palbociclib, Nu 6140 |
| | | Phospho-Ser780 | MCF-7 | 17β-Estradiol, EGF | Palbociclib, Nu 6140 |
| 7 | S6 ribosomal protein | Phospho-Ser235/236 | HEK293 | Insulin, Serum | Rapamycin |
| | | Phospho-Ser240/244 | HEK293 | Insulin, Serum | Rapamycin |
| 8 | MEK1/2 | Phospho-Ser217/221 | THP-1 | PMA, EGF | n.d |
| | | Phospho-Ser298 | Hela, HEK293 | PMA, EGF | n.d |
| 9 | ER | Total[a] | MCF-7 | N/A | Fulvestrant |
| 10 | β-Catenin | Total[a] | HEK293 | Wnt3a | JW67 |
| 11 | CREB | Phospho-Ser133 | HEK293 | Forskolin | Staurosporine |
| 12 | p38 MAPK | Phospho-Thr180/Tyr182 | A431 | Anisomycin | n.d |
| 13 | 4E-BP1 | Phospho-Ser65 | HEK293, MCF7 | Insulin | Rapamycin |

[a]Protein targets whose degradation was detected

treatment. Interestingly, both cell types showed a similar pattern of IκBα phosphorylation and degradation with just a slight difference in the magnitude of phosphorylation (Fig. 4d, e).

**Detection of important targets in other signaling pathways**. Because this bioluminescent immunoassay relies on NanoBiT-labeled secondary antibodies, it would be capable to detect virtually any target protein provided two primary antibodies specific to the target generated in separate species, such as rabbit and mouse, are used. To evaluate the utility of the bioluminescent immunoassay in detecting other targets effectively and put to the test its universality, we tested many different phospho and total protein targets. For each target, optimal primary antibody pairs and their concentrations were first identified following the procedure described in Fig. 2a. Second, using the NanoBiT detecting antibodies with optimal primary antibodies for each target, we measured total or phosphoprotein level changes in response to known pathway activators or inhibitors. A list of 13 protein targets from different signaling pathways and the detection of their phosphorylated and/or total level is shown in Table 1. In this report more in-depth analysis is presented for four of these targets, IκBα, p65, AKT, and STAT3 that could be considered as nodes for three signaling pathways (TNFα/NF-κB, Insulin/AKT, and IL-6/JAK/STAT pathways).

In the NF-κB pathway, the p65 protein is another important player that is phosphorylated and translocated into the nucleus to increase NF-κB target gene expression after its inhibitory regulator IκBα is degraded in response to cytokine treatment[21]. Using MCF-7 cells, we measured the level of phospho and total p65 in response to activation and inhibition of the pathway with TNFα and IKK16, respectively. Consistent with IκBα results, phosphorylation of p65 on serine 536 was increased after TNFα treatment and the increased phosphorylation signal was inhibited by IKK16 treatment (Fig. 5a). In contrast, the level of p65 was not affected by TNFα nor IKK16 treatment (Fig. 5d).

AKT is a serine/threonine kinase known to be an important node in cell signaling downstream of many cellular stimuli such as

growth factors[22]. AKT is activated through lipid kinase-dependent phosphorylation on two sites, Serine 473 and Threonine 308, to drive multiple cellular processes such as insulin-dependent metabolic responses, proliferation, and cell survival. Using the NanoBiT immunoassay, we tested activation and deactivation of the insulin signaling pathway in MCF-7 cells through measurement of AKT serine 473 phosphorylation in response to insulin and to a known PI3 kinase inhibitor. AKT phosphorylation in cells stimulated with insulin increased dramatically, more than 20-fold over the level detected in non-treated cells (Fig. 5b). Moreover, this increase in AKT phosphorylation was suppressed in cells treated with the PI3K inhibitor LY294002. Conversely, the total level of AKT protein stayed constant after insulin or LY294002 treatments (Fig. 5e) suggesting that the treatments did not interfere with the general behavior of the cells and that the data obtained with the NanoBiT phospho AKT immunoassay was specific to the change in the AKT S473 phosphorylation level. AKT is phosphorylated in response to different pathway activations by two kinases, mTOR and PDK1[22]. The data shown with the LY294002 compound, which inhibited PI3 Kinase upstream of AKT but not the kinase responsible for direct phosphorylation of S473, reinforce the concept of a signaling node whose phosphorylation or level (degradation) can be used as a reporter of any signaling event occurring upstream of the node.

The JAK/STAT pathway is prominent in cellular responses to a variety of cytokines and growth factors. JAK (Janus Kinases) activation upon stimulation of cells by cytokines orchestrates diverse immune and inflammatory responses[23]. One of the important nodes in JAK/STAT signaling is the STAT3 protein which is phosphorylated by activated JAK on Tyrosine 705 after IL-6 treatment[24]. Using a NanoBiT STAT3 immunoassay, we measured this phosphorylation in A431 cells. In Fig. 5c, we show that STAT3 phosphorylation in cells stimulated with IL-6 was increased and this increase was suppressed in cells treated with the JAK1/JAK2 selective inhibitor drug, Ruxolitinib. On the other hand, the total level of STAT3 did not change by any of the treatments (Fig. 5f).

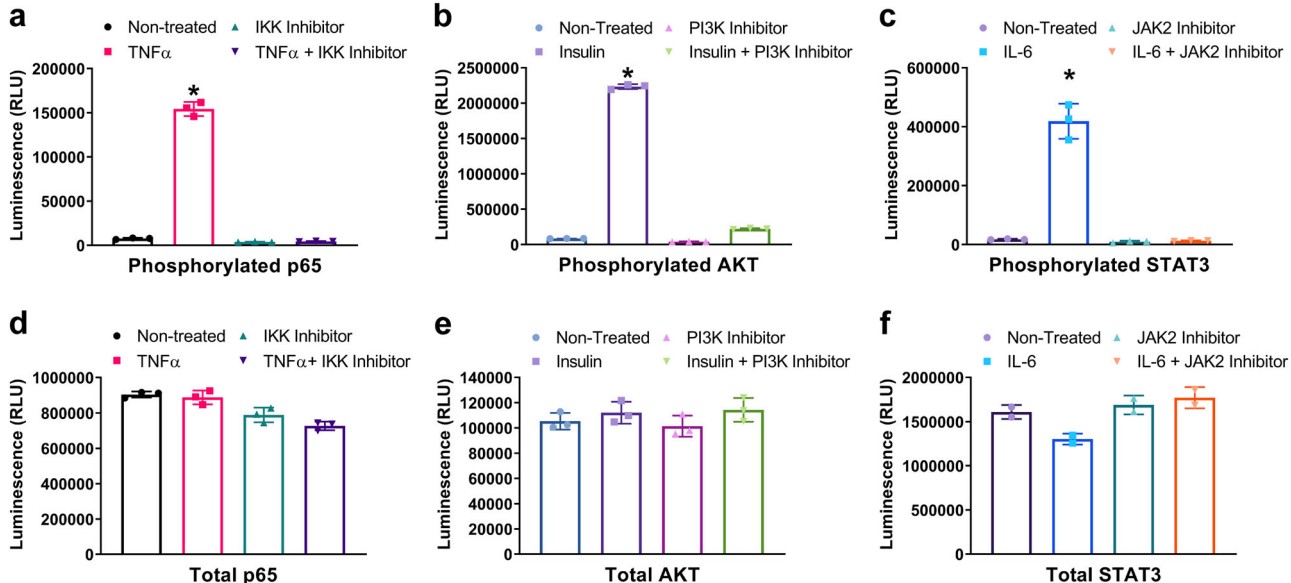

**Fig. 5 Detection of total and phosphorylated target proteins in key signaling pathways using the NanoBiT cell-based immunoassay. a, d** In all, 50,000 seeded MCF-7 cells were left untreated or pretreated with IKK16 (10 μM, 1 h) and then were untreated or treated with TNFα (50 ng per ml, 10 min) before phosphorylated p65 (**a**) or total p65 (**d**) levels were measured. **b, e** 50,000 seeded MCF-7 cells were incubated at 37 °C in a 5% CO$_2$ overnight without serum. The cells were untreated or pretreated with LY294002 (20 μM, 1 h) and then were treated with insulin (1 μM, 10 min). Phosphorylated AKT (**b**) or total AKT (**e**) were measured by NanoBiT cell-based immunoassays. **c, f** 50,000 seeded A431 cells were incubated at 37 °C in a 5% CO$_2$ overnight without serum. The cells were untreated or pretreated with Ruxolitinib (1 μM, 1 h) and then were treated or not with IL-6 (10 ng per ml, 30 min) before phosphorylated STAT3 (**c**) or total STAT3 (**f**) were measured by NanoBiT cell-based immunoassays. Results are presented as means ± S.E.M. ($n = 3$ technical replicates, the data are representative of two or more experiments). *$P < 0.001$ relative to Non-treated.

Figure 5 and Table 1 speak to the scalability of the NanoBiT cell-based immunoassay to measure diverse phosphorylation events and target protein levels in virtually any cell line, provided that two primary antibodies are available (e.g. one raised in rabbit and the other in mouse) that recognize separate epitopes on a single target protein.

**NanoBiT cell-based immunoassay application to drug discovery.** Signaling pathway regulation is tightly controlled in the cell, whereas dysregulation is central to diseases including cancer and immune diseases. Therefore, normalizing signaling pathway activity with small or large molecule drugs is a long-standing approach to drug discovery. Cell-based assays are often used to screen for and characterize drugs against specific disease validated targets and we expected the NanoBiT immunoassays would be well suited for screening against native target proteins in unmodified cells. As an example, we measured the inhibition of TNFα induced degradation of IκBα by the IKK complex inhibitor IKK16 and generated the corresponding IC$_{50}$ value. As shown in Fig. 6a, IκBα degradation in MCF-7 cells treated with TNFα was inhibited by IKK16 in a dose-dependent fashion with an IC$_{50}$ of 1.44 μM, which is similar to what was previously reported[25]. When IKK16 was tested on human primary breast tumor epithelial cells, a similar inhibition pattern of degradation was observed with an IC$_{50}$ value of 1.82 μM. We also tested the effects of well-known inhibitors on two other key signaling pathways. Increased phosphorylation of AKT after insulin treatment was inhibited by the PI3K inhibitor LY294002 in a dose-dependent fashion with an IC$_{50}$ of 8.89 μM in MCF-7 cells (Fig. 6b). Also, increased phosphorylation of STAT3 in response to IL-6 activation in A431 cells was inhibited by the JAK2 specific inhibitor Ruxolitinib with an IC$_{50}$ of 46 nM (Fig. 6c). The IC$_{50}$ values generated in these experiments were all in concordance with what was previously reported[26,27] demonstrating that these bioluminescence-based

assays are generally useful for cell-based identification of pathway activation or target phosphorylation inhibitors.

Compound selectivity is crucial during drug development to prevent any side effects of the drugs once used as a therapy. To see if the NanoBiT cell-based immunoassay can be used to profile drugs and assess their selectivity towards different node kinases belonging to the same family, STAT3 phosphorylation in response to IL-6 was studied in the presence of JAK1/2 and JAK3 kinase selective inhibitors. Ruxolitinib, a JAK1/2 selective inhibitor, effectively inhibited the JAK2 mediated phosphorylation of STAT3 after IL-6 treatment with high potency (IC$_{50}$ = 46 nM) (Fig. 6c). JAK3 kinase is not known to mediate phosphorylation of STAT3 on tyrosine 705 in response to IL-6 treatment. Therefore, when FM-381, which is more selective for JAK3 over the other JAK kinase family members (reported IC$_{50}$ of 127 pM for JAK3)[28] was used in comparison to Ruxolitinib, it inhibited STAT3 phosphorylation in response to IL-6 treatment with less potency (IC$_{50}$ of 930 nM Vs 46 nM for Ruxolitinib). This result shows that NanoBiT cell-based immunoassay can be used to screen compounds for selectivity in inhibiting signaling pathways driven by different kinases belonging to the same kinase family.

In recent years, many drug discovery efforts have focused on large molecule inhibitors such as monoclonal antibodies. These biologic drugs typically bind to cellular receptors or their ligands to prevent their interaction and block pathway activation. To demonstrate the utility of NanoBiT immunoassays for large molecule drug screening, we studied the IL-6/IL-6 receptor interaction that triggers STAT3 phosphorylation by JAK2. Figure 6d shows that an anti-IL-6 antibody inhibited STAT3 phosphorylation in a dose-dependent manner, presumably by interfering with IL-6 cytokine binding to the IL-6 receptor. Similarly, Tocilizumab, which is a humanized monoclonal antibody against the IL-6 receptor and an approved immunosuppressive drug for the treatment of rheumatoid arthritis[29],

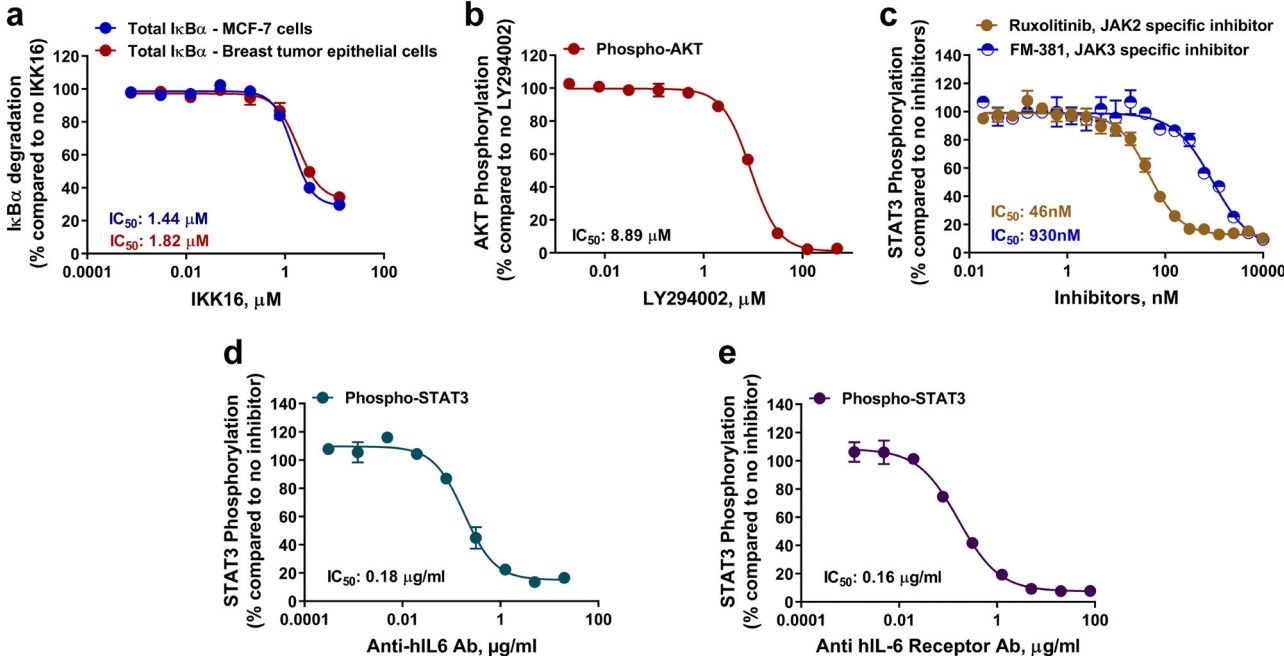

**Fig. 6 Effect of small and large molecules on phosphorylation or degradation of node proteins in key signaling pathways. a** In all, 50,000 seeded MCF-7 or human breast tumor epithelial cells were pretreated with various concentration of IKK16 for 1 h and then untreated or treated with TNFα (50 ng per ml, 30 min) before total IκBα was measured by NanoBiT cell-based immunoassays. **b** In all, 50,000 seeded MCF-7 cells were incubated at 37 °C in a 5% $CO_2$ overnight without serum. The cells were pretreated with various concentration of LY294002 for 1 h and then untreated or treated with insulin (1 μM, 10 min). Phosphorylated AKT was measured by NanoBiT p-AKT cell-based immunoassay as described. **c–e** 50,000 seeded A431 cells were incubated at 37 °C in a 5% $CO_2$ overnight without serum. The cells were pretreated with various concentration of Ruxolitinib or FM-381 (**c**), anti IL-6 antibody (**d**), or anti IL-6 receptor Tocilizumab (**e**) for 1 h and then were untreated or treated with IL-6 (10 ng per ml, 30 min). Phosphorylated STAT3 was measured by NanoBiT pSTAT3 cell-based immunoassay. Results are presented as means ± S.E.M. (n = 3 or four technical replicates, the data are representative of two or more experiments).

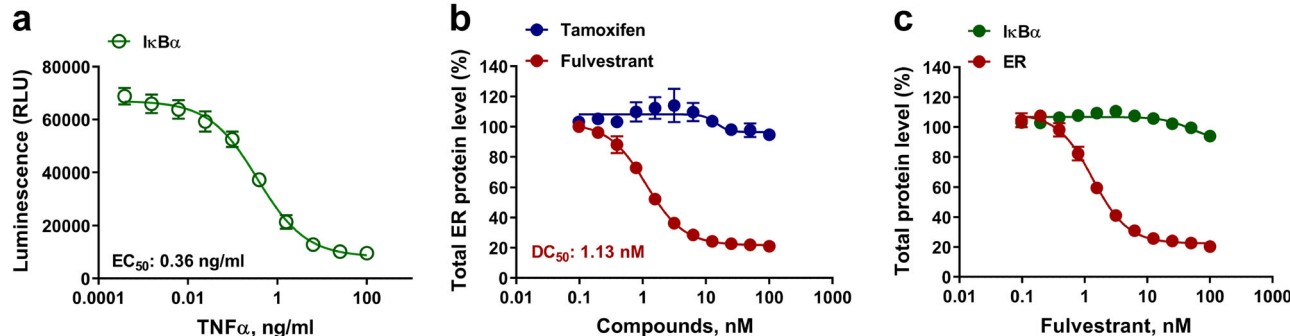

**Fig. 7 Detection of natural and small molecule targeted degradation of proteins using NanoBiT cell-based immunoassay. a** In all, 50,000 seeded MCF-7 cells were treated with various concentration of TNFα for 30 min, before total IκBα level was measured. **b** In all, 50,000 seeded MCF-7 cells were incubated at 37 °C in a 5% $CO_2$ overnight without serum. The cells were treated with various concentration of Fulvestrant or Tamoxifen for 4 h. Total ER level was measured by NanoBiT cell-based immunoassay using ER primary antibodies described in Material and Methods. **c** In all, 50,000 seeded MCF-7 cells were incubated at 37 °C in a 5% $CO_2$ overnight without serum. The cells were untreated or treated with various concentration of Fulvestrant for 4 h. Total ER or IκBα level were measured by NanoBiT cell-based immunoassays. Results are presented as means ± S.E.M. (n = 3 technical replicates, the data are representative of two or more experiments).

inhibited JAK2-STAT3 pathway activation in a dose response manner (Fig. 6e).

Recently, targeted protein degradation has emerged as a therapeutic strategy. The new class of drugs, called protein degraders, represent a novel way to decrease the level of protein targets considered "undruggable" such as transcription factors or proteins with no catalytic activity. Many variants of protein degraders exist but in general they all follow the same principle of action which is binding disease validated targets and directing them to the cell's Ubiquitin/Proteasome system to

be degraded[30]. With a NanoBiT immunoassay we measured IκBα protein degradation as a natural response to cytokines treatment (Fig. 7a). To show drug-induced degradation with our assay system we applied Fulvestrant to cells that express estrogen receptors (ER). Fulvestrant is an ER antagonist that binds ER and triggers its dimer dissociation resulting in instability, which in turn targets it to the proteasome for degradation[31–33]. Fulvestrant was titrated and its effect on ER protein level was detected in MCF-7 cells using NanoBiT detecting antibodies and a validated pair of anti-ER primary

antibodies. Tamoxifen, a well-known ER signaling inhibitor that also binds ER, but does not trigger its degradation, was used as a negative control. As predicted, Fulvestrant induced ER degradation in a dose-dependent manner, lowering the total ER protein level by over 80% with a degradation constant $DC_{50}$ value of 1.13 nM (Fig. 7b). In contrast, Tamoxifen did not have any effect on the detected level of ER protein. To test Fulvestrant selectivity for ER and to determine if the compound had a toxic effect on cells during the experimental time frame, we repeated the experiment using the same compound titration, but measured IκBα instead. The results showed that Fulvestrant was indeed a specific ER degrader as it did not affect the IκBα protein level. This also infers the lack of cytotoxicity because cell death would have caused a decrease in the IκBα level proportionate to the decrease in cell number (Fig. 7c). This control experiment further shows that Fulvestrant did not non-specifically affect NanoBiT immunoassay luminescence. Taken together, these results showed that the NanoBiT cell-based immunoassay is an effective drug discovery tool that detects the modulation of protein target phosphorylation in response to chemical compounds or biologics and can identify drugs that alter target protein levels.

## Discussion

We developed a bioluminescent cell-based immunoassay system and demonstrated its utility in easily detecting the phosphorylation and total level of various protein targets. The NanoBiT immunoassay is homogeneous and requires only three addition steps and <2 h to perform. This assay can be adopted for a range of targets provided an appropriate pair of primary antibodies against the target is available. We measured the level of over 20 target proteins and/or their phosphorylation in response to signaling cues. By selecting specific targets in different signaling pathways, we demonstrated that the changes in these can be considered as nodes of the pathways that can report on the status of the pathway regulation. We showed that some nodes can report on the biological activity of multiple signaling pathways as well. The fact that this bioluminescent immunoassay is simple to use can increase the level of intricacy that can be analyzed in one experiment. Furthermore, the NanoBiT cell-based immunoassay was tested in many cell types including primary cells, demonstrating its value in studying real endogenous biology where no modification of cell machinery is required. This has advantages when working with primary cells such as patient derived normal or tumor cells. In addition, the bioluminescent immunoassays are useful for pathway inhibitor studies, drug screening and drug profiling in cell-based mode. Furthermore, these robust assays serve as bioassays to identify and characterize drugs of diverse structures including antibody therapeutics and new chemical compounds such as protein degraders.

A variety of methods have been developed over the years to detect the level of proteins or PTMs in cell lysates, including heterogenous ELISA and western blot-based assays, and homogeneous HTRF- or AlphaLISA-based methods. Some of the drawbacks of those methods are low assay throughput and the requirement of multiple wash steps (Supplementary Fig. 2), lengthy protocols and risk of assay interference due to chemical compound fluorescence. To our knowledge this is the first report on the use of bioluminescence in a homogeneous targeted immunodetection scheme. This provides the possibility of target detection in small cell numbers due to the sensitivity provided by signal amplification and more resistance to drug interference.

In summary, the results presented here demonstrate the usefulness of a bioluminescent cell-based immunoassay as a new tool

for cell biology studies and drug discovery. This approach could have a significant impact on streamlining signaling pathway analyses, studies of kinase (or other PTM enzymes) activities and regulation and identifying chemical and biological drugs in relevant unmodified cell types.

## Methods

**Reagents.** The following antibodies were purchased from Cell Signaling Technology Inc. and were used at 150 ng/ml: Rabbit anti-phospho-IκBα (#2859) (indicated as #1 in Fig. 2a), Mouse anti-phospho-IκBα (#9246) (indicated as #2 in Fig. 2a), Mouse anti-IκBα (#4814) (indicated as #1 in Fig. 2a), Mouse anti-IκBα (#9247) (indicated as #2 in Fig. 2a), Rabbit anti-IκBα (#4812) (indicated as #3 in Fig. 2a), Mouse anti-phospho-p65 (#13346), Rabbit anti-p65 (#8242), Mouse anti-p65 (#6956), Mouse anti-phospho-AKT (#4051), Rabbit anti-AKT (#4691), Mouse anti-AKT (#2966), Rabbit anti-phospho-STAT3 (#9145), Mouse anti-STAT3 (#9139), and Rabbit anti-STAT3 (#8768). Rabbit anti-ER (#MA5-14501) and Mouse anti-ER (#MA5-13191) antibodies were purchased from ThermoScientific and were used at 150 ng/ml. Recombinant human TNFα (#654205) was purchased from CalBiochem. Insulin (#I-0516) was purchased from Sigma-Aldrich. Recombinant human Interleukin-6 Protein (#IL006) and Interleukin-1β (#IL038) Protein were purchased from EMD Millipore. Halt Protease and Phosphatase inhibitor cocktail (#78445) was purchased from ThermoScientific. Ruxolitinib (#R-6600) was purchased from LC Laboratories. FM-381 (#S8541) was purchased from Selleckchem. Anti-human IL-6 antibody (#AF-206-NA) was purchased from R&D Systems and was used in a titration from 0.76 ng/ml–20 μg/ml. ACTEMRA® (tocilizumab, # NDC 50242-136-01) was purchased from Genentech and was used in a titration from 0.0012 to 80 μg/ml. Tamoxifen (#6342) and Fulvestrant (#1047), were purchased from Tocris. MG132 (#G932) and LY294002 (#V1201) were obtained from Promega.

The NanoBiT cell-based immunoassay components include Nano-Glo® Luciferase Assay Substrate (#N113), digitonin (#G944), 10x Immunoassay buffer (10 mg per ml BSA in 10X TBS), antibody dilution buffer (10 mM HEPES, 150 mM NaCl, 0.1% BSA, 50% Glycerol) and small or large BiT labelled secondary antibodies. To create NanoBiT-labeled secondary antibodies, the SmBiT and LgBiT recombinant proteins were expressed in E. coli as genetic fusions with HaloTag by combining existing NanoBiT and HaloTag sequences (Promega). The NanoBiT to antibody conjugation protocol was modified from Nath et. al.[34] In brief, anti-mouse or anti-rabbit secondary antibodies were first labeled with amine reactive HaloTag Succinimidyl Ester (O4) Ligand (Promega). Then, HaloTag-SmBiT or HaloTag-LgBiT recombinant fusion proteins were covalently attached to the HaloTag ligand activated antibodies to form the NanoBiT detecting antibodies. Note that the HaloTag ligand forms a covalent bond with the HaloTag protein, which in turn links either SmBiT or LgBiT to a particular antibody. The NanoBiT detecting antibodies were diluted to approximately 250 μg per ml in the antibody dilution buffer.

**Cell culture.** All cell lines used are from ATCC. MCF-7, A431, HEK293, and HeLa cells were cultured in DMEM medium (Gibco #11995-065) supplemented with 10% (v/v) fetal bovine serum (Seradigm #1500-050), and antibiotics (Gibco #15140-122, 100 U per ml penicillin and 100 μg per ml streptomycin). Ramos-RA1 and THP-1 cells were cultured in RPMI 1640 medium (Gibco #22400-089) supplemented with 10% (v/v) fetal bovine serum (Seradigm #1500-050), and antibiotics (Gibco #15140-122, 100 U per ml penicillin and 100 μg per ml streptomycin). Human Primary Mammary Epithelial Cells (#H-6035), Human Breast Tumor Epithelial Cells (#HC-6035), Complete Human Epithelial Cell Medium (#H6621), and Gelatin-Based Coating Solution (#H6950) were purchased from Cell Biologics. The cells were cultured according to the manufacturer's procedure. All cells were grown at 37 °C in a 5% $CO_2$ humidified incubator and passaged before reaching 80% cell confluency. Prior to each experiment, cells were counted by combining 30 μl of cell suspension with 30 μl of trypan blue. After gentle pipetting up and down to mix the cells and the dye, the cell number was determined using Bio-Rad TC 20 cell counter.

**NanoBiT cell-based immunoassay protocol.** Prior to NanoBiT Cell-Based Immunoassay, 50,000 cells were seeded in 160 μl of complete growth media per well of a 96-well plate and incubated overnight. The cells were checked under microscope and it was determined that 50,000 cells per well in 96-well plate is under 90% confluency for all cells tested. If necessary, the cell culture media were replaced with serum-free media after the cells were stably adherent on the plate and incubated overnight. To treat cells with activators or inhibitors, medium was changed to 30 μl of medium containing inhibitors. If activation of a specific pathway is needed, the cells were stimulated with 10 μl of medium containing activators. To lyse the cells, 10 μl of lysis buffer containing 0.1% digitonin diluted in 1X immunoassay buffer with protease and phosphatase inhibitors was added and the plates were mixed vigorously for 20 min. Then, 50 μl of an antibody mix containing two primary antibodies against the target protein and NanoBiT detecting antibodies (0.3 μl each per well), diluted in 1X immunoassay buffer was added to the lysates. The plates were incubated at 23 °C for 90 min, followed by the

addition of 25 µl of Nano-Glo® reagent consisting of Nano-Glo® Luciferase Assay Substrate mixed with 1X immunoassay buffer. Luminescence was measured using a plate-reading luminometer.

**Optimization of IκBα primary antibody pairs**. In all, 50,000 MCF-7 cells were seeded in 160 µl complete growth medium per well of a 96-well plate and incubated overnight. Half of the cells were treated with 50 ng per ml TNFα and half with vehicle. Cells were lysed as described above and a mixture of NanoBiT secondary antibodies with primary antibodies at the matrix concentrations described in Fig. 2a were added. After 90 min incubation, the Nano-Glo® reagent was added to the reactions as described above and luminescence was measured using a plate-reading luminometer.

**Comparison of western blot versus NanoBiT immunoassay**. In order to compare the NanoBiT immunoassay with Western blot analysis, MCF-7 cells were seeded at different concentrations per well as indicated in two 96-well plates and incubated overnight. In each plate, half of the cells were treated with 50 ng per ml TNFα to detect phospho IκBα and half with vehicle to detect total IκBα protein. After treatment, the samples in the first plate were analyzed with NanoBiT immunoassay as described above. For Western blot analysis, the medium was removed, and SDS-PAGE sample buffer was added to samples. Samples were transferred to 1.5 ml microcentrifuge tubes, heated at 98 °C for 3 min, and western blot analysis was performed as previously described[35].

**Pathway inhibition studies**. Single dose or dose response inhibitor treatments were performed prior to activation of the pathways as follows. For IκBα, the cells were treated in 30 µl medium with IKK16 or MG132 for 60 min before the addition of 10 µl of DMEM containing appropriate concentration of TNFα. To measure phosphorylation of AKT or STAT3, the corresponding NanoBiT immunoassays were developed with MCF-7 or A431 cells, respectively, according the same scheme used to develop the IκBα assay using the primary antibodies for AKT and STAT3 described in the reagents section. For the AKT assay, MCF-7 cells were incubated overnight in phenol-red free DMEM medium (Gibco #21063-029) without serum. Treatment with LY294002 was performed in 30 µl medium and cells were incubated for 60 min before the addition of 1 µM Insulin. To measure STAT3 phosphorylation, A431 cells were incubated overnight in DMEM medium without serum. Cells were pre-incubated with Ruxolitinib, FM-381, Anti-human IL-6 antibody, or tocilizumab for 60 min before 10 ng per ml IL-6 was added to the cells. After addition of the stimulators, the cells were incubated for additional times as follows. IκBα phosphorylation and degradation analyses required both time courses and single time points as described in figures. For AKT and STAT3, cells were treated with Insulin or IL-6 for 10 min and 30 min, respectively. The NanoBiT immunoassay protocol was followed as described above.

**Signal detection and data analysis**. All 96-well assay plates were read using a GloMax® 96 Microplate Luminometer from Promega. The instrument was set to 0.5 s integration time. In order to obtain net change of signal, the reagent background values (signal from the well containing all reagent except primary antibodies) were subtracted from the sample values. To plot and analyze the data, Microsoft Excel and GraphPad Prism® Software were used. $IC_{50}$ values were determined by using a nonlinear regression fit to a sigmoidal dose response (variable slope).

**Statistics and reproducibility**. Values are presented as means ± S.E.M. with the indicated number of independent experiments. The statistical significance of differences between groups was determined by the Student's $t$ test (two-tailed). $p$ values < 0.05 were considered statistically significant.

**Reporting summary**. Further information on research design is available in the Nature Research Reporting Summary linked to this article.

## Data availability
The authors declare that the main data supporting the findings of this study are available within the article and its Supplementary Information files. Source data are available in Supplementary Data 1.

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

## Acknowledgements

We thank James J. Cali for his critical reading of the paper and Nidhi Nath for invaluable input and support throughout this study.

## Author contributions

B.H. and H.Z. generated idea for the technology. B.H., S.A.G, and H.Z. refined the idea. B.H. and H.Z. designed experiments, B.H. and L.E. performed experiments, and B.H, S.A.G, and H.Z. wrote the paper.

## Competing interests

All authors are employees of Promega Corporation.
