## [Peer Review File · Communications Biology]

Reviewers' comments:

Reviewer #1 (Remarks to the Author):

Hwang and colleague showed a novel and sensitive assay for endogenous protein using NanoBiT-tagged antibodies. This study is interesting; however, there are serious concerns about experimental conditions and results. Therefore, several experiments and also some modifications are needed to accept these conclusions.

1, Authors tried several experiments using this NanoBiT-based assay; however, the results of conventional western blots are shown only in Fig. 3CD. Authors should perform the expression analysis of each protein using conventional western blotting throughout this manuscript.

2, In Fig, 2, authors tested the several experimental conditions and determined that 50.000 cells/ well is optimal. It seems to be too confluent. Under the confluent condition, cellular viability and responses will change.

3, In this NanoBiT-based assay, the digitonin-treated cells (not fixed) were incubated with 1st or 2nd Ab for the indicated time. Authors should perform the western blotting to verify the each protein expression and its phosphorylation status dose not change before and after these labeling processes.

4, Authors describe the WB signals of total IκBα reached a plateau (Fig. 3d); however, it feels that this is too long exposure. Authors should re-blot and quantify a short exposed image.

5, In Fig. 4b, MG132-treatment did not influence total IκBα expression. Is it correct? Authors should detect the IκBα expression as well as other experiments by western blotting.

Reviewer #2 (Remarks to the Author):

This manuscript reported a novel homogeneous cell-based bioluminescent immunoassay for the detection of the phosphorylation and total level of native target proteins in cells. The assay was simple, fast, and applicable to various target proteins in different types of cells. It would be very useful in the study on signaling pathway in cells and relevant drug screening. The manuscript was acceptable for publication after minor revisions.

Specific comments:

1. The lower limit of detection of the bioluminescent immunoassay for some typical target proteins may be measured and added to the manuscript.
2. Will the binding of primary antibody to the phosphorylated target protein affect the latter's degradation speed?
3. The method for determination of the cell number should be described? The standard deviation of the cell number data needs to be provided.

Response: Reviewers comments answered below.

Reviewer #1 (Remarks to the Author):

1, Authors tried several experiments using this NanoBiT-based assay; however, the results of conventional western blots are shown only in Fig. 3CD. Authors should perform the expression analysis of each protein using conventional western blotting throughout this manuscript.

We presented two examples of western data (p-IkB and Total IkB) as representative for comparison to the NanoBiT cell-based immunoassay in order to benchmark the different features and performance of the new assay against western blot. The new assay presented here generates similar data as other well-known technologies, but in a simple, easy to use, and homogenous format making it amenable to high through put assays commonly used for drug discovery research programs.

2, In Fig, 2, authors tested the several experimental conditions and determined that 50.000 cells/ well is optimal. It seems to be too confluent. Under the confluent condition, cellular viability and responses will change.

The assay is performed in 96 well plates and based on microscope analysis, the 50,000 cells seeded are non confluent. When tested lower cell number/well for all the targets, the signal fold changes upon activations were similar. For clarification, we added a text to this effect in the revised manuscript version.

3, In this NanoBiT-based assay, the digitonin-treated cells (not fixed) were incubated with 1st or 2nd Ab for the indicated time. Authors should perform the western blotting to verify the each protein expression and its phosphorylation status dose not change before and after these labeling processes.

The NanoBiT cell-based immunoassay is performed without any labeling process. This immunoassay is an end point homogeneous assay which means the lysis, antibody binding to the target and the luminescence generation steps all happen in solution without transfer nor washing steps. The optimized Digitonin treatment lysed the cells completely to create a homogeneous lysate that immediately terminates any biochemical reaction including protein expression and phosphorylation. During the development of this assay, we compared the protein level and phosphorylation using different cell disruption protocols such as sonication, and other lysis buffers. We obtained similar results using sonication and digitonin treatment, but we chose the latter because of its compatibility with high through put formats which is a very desirable feature for drug discovery programs. This was also the case when comparing this technology and western blotting since the latter is limited to few samples at a time. Therefore, we don't expect that assay components (antibodies) will change the protein level nor phosphorylation state of the targets studied since these manipulations were included in the various cell lysis protocols tested. For clarification, we added a text to this effect in the revised manuscript version.

4, Authors describe the WB signals of total I κ B α reached a plateau (Fig. 3d); however, it feels that this is too long exposure. Authors should re-blot and quantify a short exposed image.

The authors respectfully disagree with this assessment. The exposure time was the shortest possible (10s) and the level of I κ B detected indeed reached a plateau at higher cell numbers.

5, In Fig. 4b, MG132-treatment did not influence total I κ B α expression. Is it correct? Authors should detect the I κ B α expression as well as other experiments by western blotting.

This experiment is designed to show the effect of TNF on the I κ B level during a short time course. Cells were treated for 1 hour with or without MG132 to block the proteasome and then TNF was added to trigger I κ B degradation. We don't believe that MG132 indirectly or directly influence I κ B expression. However, it blocks the proteasome and indirectly prevents I κ B from degrading in response to NF- κ B pathway activation by TNF and that's is what was reported in the figure.

Reviewer #2 (Remarks to the Author):

1. The lower limit of detection of the bioluminescent immunoassay for some typical target proteins may be measured and added to the manuscript.

The sensitivity reported is based on lowest number of cells that can be used in the assay and still detect the changes in biology studied. We understand that when comparing two detection assays, the sensitivity is commonly better compared by looking at the lower limit of detection of the target. However, this is true for detecting a titrated analyte in a buffer. It is much difficult to do when comparing the target level in the cells. This is why we chose to do that using cell titration. Therefore, we believe that showing the detection of these two targets (I κ B and p-I κ B) with different levels of abundance using a range of cell number reflects the comparison of limit of detection needed. Moreover, the manuscript focuses more on the simplicity of the NanoBiT method to decipher signaling biology and this new method has the sensitivity required for that.

2. Will the binding of primary antibody to the phosphorylated target protein affect the latter's degradation speed?

Answered above in response to reviewer 1, comment 3.

3. The method for determination of the cell number should be described? The standard deviation of the cell number data needs to be provided.

For clarification, we added a text describing how we determine cell numbers to seed per well in the revised manuscript version. Standard deviation of all the data described in figures is plotted as error bars in every figure.